# MicroRNAs in Neuroinflammation: Implications in Disease Pathogenesis, Biomarker Discovery and Therapeutic Applications

**DOI:** 10.3390/ncrna5020035

**Published:** 2019-04-24

**Authors:** Jessy A. Slota, Stephanie A. Booth

**Affiliations:** 1Prion Diseases Section, Public Health Agency of Canada, National Microbiology Laboratory, 1015 Arlington St., Winnipeg, MB R3E 3R2, Canada; slotaj@myumanitoba.ca; 2Department of Medical Microbiology and Infectious Diseases, Faculty of Health Sciences, University of Manitoba, 730 William Ave., Winnipeg, MB R3E 0W3, Canada

**Keywords:** microRNA, neuroinflammation, neurodegeneration, central nervous system, immunity, biomarkers, therapeutics

## Abstract

The central nervous system can respond to threat via the induction of an inflammatory response. Under normal circumstances this response is tightly controlled, however uncontrolled neuroinflammation is a hallmark of many neurological disorders. MicroRNAs are small non-coding RNA molecules that are important for regulating many cellular processes. The ability of microRNAs to modulate inflammatory signaling is an area of ongoing research, which has gained much attention in recent years. MicroRNAs may either promote or restrict inflammatory signaling, and either exacerbate or ameliorate the pathological consequences of excessive neuroinflammation. The aim of this review is to summarize the mode of regulation for several important and well-studied microRNAs in the context of neuroinflammation, including miR-155, miR-146a, miR-124, miR-21 and let-7. Furthermore, the pathological consequences of miRNA deregulation during disorders that feature neuroinflammation are discussed, including Multiple Sclerosis, Alzheimer’s disease, Parkinson’s disease, Prion diseases, Japanese encephalitis, Herpes encephalitis, ischemic stroke and traumatic brain injury. There has also been considerable interest in the use of altered microRNA signatures as biomarkers for these disorders. The ability to modulate microRNA expression may even serve as the basis for future therapeutic strategies to help treat pathological neuroinflammation.

## 1. Brain Immunity and Neuroinflammation

The central nervous system (CNS) is traditionally thought of as an immune-privileged site, separated from peripheral immune cells that are unable to cross the blood brain barrier (BBB) under normal conditions [1]. In response to threat, resident glial cells including microglia and astrocytes become activated and induce an inflammatory response through pro-inflammatory cytokines, chemokines, secondary messengers and reactive oxygen species (ROS) [2]. This response, termed neuroinflammation, may be beneficial, by protecting the brain from pathogens and neurotoxic agents and promoting tissue repair processes [3,4]. Conversely, uncontrolled neuroinflammation can cause pathogenic tissue damage within the CNS through elevated glial cell activation, BBB permeability and infiltration of peripheral immune cells [5]. These changes ultimately lead to increased levels of pro-inflammatory mediators within the CNS, many of which are neurotoxic and can induce neurodegeneration [5]. Although neuroinflammation is a component of many neurological disorders, it is often difficult to elucidate the role played by these processes as there are both beneficial and harmful aspects of neuroinflammation [4]. Indeed, a balance must be struck between protecting the brain from threats, while preventing damage from uncontrolled inflammation. Therefore, understanding the cellular and molecular players that control neuroinflammation is an area of ongoing interest which may shed light on the pathogenic mechanisms behind neurological disease and lead to the discovery of novel therapeutics.

Among the principal CNS resident cells that mediate neuroinflammation are microglia, a specialized long-term resident macrophage that develops early during embryogenesis from myeloid precursor cells [6]. Under normal conditions within the CNS, microglia are traditionally thought to exist in a “quiescent” or “resting” state in which they continuously scan the surrounding microenvironment and help maintain brain homeostasis through roles in synapse organization, removal of debris by phagocytosis and release of neurotrophic factors [7]. Microglia express an array of pattern recognition receptors, cytokine receptors and neuronal receptors which can respond to a variety of pathogen associated molecular patterns (PAMPs), danger associated molecular patterns (DAMPs) and other molecular signatures, triggering signaling that leads to microglial activation [8]. Activation of microglia is accompanied by morphological changes; quiescent microglia are characterized by a ramified morphology, but upon activation they take on an amoeboid form which permits motility and phagocytosis [7]. Depending on the nature of the signals that lead to their activation, microglia may differentiate into either M1 (pro-inflammatory) or M2 (anti-inflammatory) phenotypes which resemble those seen in macrophages. M2 microglia release anti-inflammatory and protective cytokines such as IL-10, TGF-β, IL-4 and IL-13, which promotes wound healing and tissue repair [9]. M1 microglia release inflammatory mediators such as ROS, MMP-9 and pro-inflammatory cytokines such as TNFα, IL-6 and IL-1β [10]. The balance between these different microglial phenotypic states may promote inflammation or tissue repair and influence the progression of neuroinflammatory disorders [10].

Astrocytes are another functionally diverse glial cell type within the brain parenchyma that may influence the inflammatory response of the CNS to various stimuli [11]. In the normal healthy brain, astrocytes help to maintain homeostasis through metabolic support, regulation of synaptogenesis, removal of excessive neurotransmitters in the extracellular space, and through interactions with the neuronal signaling system [12]. Astrocytes also play important roles in the CNS response to stress and disease. Reactive astrogliosis, a heterogeneous spectrum of molecular and cellular changes within astrocytes following activation, is a hallmark pathology of many disorders of the CNS [13]. Neuroinflammation may be either promoted or restricted by astrocytes through a variety of mechanisms, including release of pro-inflammatory and anti-inflammatory molecules, regulation of leukocyte recruitment into the CNS and by serving as functional barriers for the CNS parenchyma [14].

Beyond the contributions made by individual cell types, neuroinflammation is regulated through complex signaling cascades between different cell types within the CNS. For instance, the neurovascular unity (NVU) is a structure formed from neurons, astrocytes, extracellular matrix and the brain microvascular endothelium which plays a pivotal role in controlling the neuroinflammatory response [15]. At the center of the NVU are astrocytes, which contact both the vascular endothelium via their astrocytic end-feet and neurons through synapses, allowing intercommunication between blood vessels and neurons within the CNS [16]. This network allows for regulation of blood flow, brain development, BBB permeability, clearance of toxic by-products and immune surveillance [17]. The NVU enables communication between the CNS and peripheral immune system through regulation of BBB permeability and leukocyte entry, which can promote neuroinflammation during a variety of brain pathologies [18].

Accordingly, it is clear that neuroinflammation is mediated through the complex interplay between cells within the CNS and the periphery. Although the brain inflammatory response is tightly controlled in the healthy brain by numerous regulatory mechanisms, under pathological states these processes may become deregulated leading to uncontrolled neuroinflammation [4]. Among the central regulators of these processes are microRNAs (miRNAs), which may either become deregulated, contributing to disease progression, or may reflect a homeostatic attempt of the CNS to prevent excessive damage and restore normal conditions.

## 2. MicroRNAs

MicroRNAs are a group of short, non-coding RNA molecules of approximately 22 nucleotides in length which post-transcriptionally regulate the expression of mRNA through the RNA interference pathway [19]. The genes encoding miRNAs are transcribed by RNA polymerase II, resulting in the primary miRNA transcript (pri-miRNA), which is usually over 1 kbp and forms a stem-loop structure [20]. However, in some cases transcription of miRNA may be carried out by RNA polymerase III [21]. The microprocessor complex is formed by the RNAse III type endonuclease Drosha and DCGR8, which cleaves the pri-miRNA within the nucleus into a small, ~65 nt RNA hairpin known as a pre-miRNA [22]. The pre-miRNA is then exported into the cytoplasm where it is cleaved near the terminal loop by Dicer, another RNAse III endonuclease [23]. This liberates a small RNA duplex containing the mature miRNA sequences which is subsequently loaded into the RNA-induced silencing complex (RISC).

The mature RISC is formed by argonaute (AGO) proteins which bind the miRNA duplex and remove the passenger miRNA strand [24]. The guide strand is selected based on the thermodynamic stability of the two ends of the RNA duplex. The strand that has a less thermodynamically stable 5’ terminus is usually selected as the guide, although this is not always the case and may be cell type specific [25]. The target mRNA molecules are recognized by sequence complementarity to the miRNA seed sequence, corresponding to positions 2–8 of the 5′-end of the guide strand [26]. Perfect complementarity between the miRNA and the target sequence usually results in cleavage of the target between nucleotides 10 and 11 by the PIWI domain of Ago [27]. Mammalian miRNAs usually bind their targets via incomplete complementarity, which leads to translational repression either through interference with the translational machinery or by targeting the mRNA for deadenylation and decay [28].

The expression of more than half of all protein coding genes is likely controlled by the over 2500 miRNAs that have been annotated in the human genome [29,30]. Each miRNA may regulate many, even hundreds of different mRNA molecules and multiple miRNAs may regulate the same mRNA [31]. MiRNAs are themselves subject to regulation at several levels including the transcriptional level and at each of the steps of their biogenesis [20]. They can also be released from cells in small membrane bound extracellular vesicles (ECV’s) that can be internalized by other cells, thereby enabling miRNAs to participate in intercellular communication [32]. Thus miRNAs represent an important regulatory system with diverse functions and unsurprisingly, miRNAs have been found to play roles in a variety of different diseases [33].

## 3. Key miRNAs Which Regulate Neuroinflammation

Neuroinflammation is among the many processes which are regulated by miRNA. A few miRNAs have been well studied in the context of neuroinflammation. Inflammatory processes may be promoted by miRNAs, such as miR-155, or suppressed by miRNAs including miR-146a, miR-124 and miR-21. Some miRNAs, such as the let-7 family may either promote or inhibit the induction of the inflammatory response.

### 3.1. miR-155

MiR-155 is a central proinflammatory mediator of the CNS which is induced within macrophages and microglia in response to NF-κB dependent TLR signaling [34,35,36,37]. Targets of miR-155 include anti-inflammatory regulators such as SOCS1 [34,36], SHIP1 [38], C/EBP-β [39] and IL13Rα1 [40]. Therefore, miR-155 lifts repression of inflammatory signaling by these suppressive factors, and accordingly contributes to the induction of neuroinflammation. Within microglia, miR-155 is able to promote inflammation by a second pathway. When miR-155 expression is induced by the transcription factor p53 it subsequently targets the transcription factor c-Maf, which induces differentiation and anti-inflammatory responses in immune cells [41]. Furthermore, p53 also suppresses c-Maf through the induction of two additional microRNAs, miR-34a and miR-145, which target Twist2, an activator of c-Maf expression [41]. By suppressing the anti-inflammatory transcription factor c-Maf, miR-155 along with miR-34a and miR-145 promote inflammatory responses.

### 3.2. miR-146a

MiR-146a is a negative regulator of inflammation which is expressed in neurons, microglia and astrocytes and is also induced by TLR signaling through NF-κB [42,43,44]. MiR-146a acts as a negative feedback regulator of NF-κB signaling by targeting components of the MyD88 signaling complex, including IRAK1 and TRAF6 [42,44,45]. MiR-146a targets other proinflammatory mediators including STAT-1 [46,47], IRF-5 [47] and CFH [44,48]. Macrophage polarization is also modulated by miR-146a through the Notch1 pathway, with miR-146a expression promoting M2 polarization [49]. Therefore miR-146a serves to diminish inflammatory signaling within the CNS following stimulation by NF-κB, helping to limit excessive neuroinflammation.

### 3.3. miR-124

MiR-124 is an anti-inflammatory miRNA that is considered to be brain specific as it is involved in the regulation of neuronal differentiation [50] and under normal conditions is highly expressed in microglia, but not peripheral macrophages [51]. However, the expression of miR-124 may be induced in monocytes and macrophages in response to the Th2 cytokines IL-4 and IL-10 [52]. The induction of miR-124 in macrophages leads to anti-inflammatory effects such as downregulation of TLR-6 and MyD88 [53], and polarization towards an M2 phenotype [52]. Similarly, miR-124 promotes a quiescent state in microglia by targeting the transcription factor C/EBP-α and its downstream target PU.1, which are master regulators of polarization in macrophages and monocytes [51]. Expression of miR-124 in microglia has also been shown to reduce inflammation through downregulation of TNF-α and MHC-II, as well as a reduction in ROS [54]. Indeed, miR-124 appears to regulate the polarization of microglia along with other key miRNA regulators of neuroinflammation. Downregulation of miR-124, miR-689 and miR-711 in microglia coincides with a transition away from resting state (M0), and other miRNA control the outcome of this activation; with increased miR-155 expression associated with pro-inflammatory (M1) polarization whereas increased miR-145 expression is associated with anti-inflammatory (M2) polarization [55]. Clearly, miR-124 is a key negative regulator of neuroinflammation by reducing inflammatory mediators and restricting microglia to an inactive state.

### 3.4. miR-21

Although miR-21 is highly expressed in a variety of active immune cells such as macrophages, mast-cells, neutrophils and T-cells [56], it plays important roles in various cell types of the CNS including microglia [57], astrocytes [58], neurons [59] and oligodendrocytes [60]. MiR-21 is another anti-inflammatory regulator that is induced by TLR signaling through MyD88 and NF-κB, and targets PDCD4, leading to downregulation of NF-κB and induction of the anti-inflammatory cytokine IL-10 [61]. Furthermore, miR-21 mediates anti-inflammatory effects in macrophages by decreasing TNF-α secretion [62,63], although it also inhibits M2 polarization by targeting STAT-3 [64]. MiR-21 is also induced in astrocytes in response to stress, and serves to impair astrocytic activation during the early stages of astrogliosis [65].

### 3.5. Let-7

Let-7 is a family of evolutionary conserved miRNA, of which there are nine in humans that generally serve as tumor suppressors and regulators of developmental processes yet remain highly expressed in adult tissues [66]. The let-7 miRNAs also serve as important modulators of neuroinflammatory processes. Let-7 miRNAs are known to promote polarization of macrophages into the anti-inflammatory M2 phenotype by targeting the C/EBP-δ transcription factor [67] as well as inhibit apoptosis and promote the M2 phenotype in microglia [68]. Let-7 is further able to modulate inflammation by targeting the cytokines IL-6 and IL-10 [69] in addition to the receptor TLR4 [70]. Along with miR-125, let-7 promotes differentiation of astrocytes by targeting negative regulators of this process within glial progenitor cells [71]. Interestingly, let-7 may also promote activation of microglia and macrophages by serving as a DAMP for TLR7 [72,73]. The let-7 family are important regulators of various inflammatory processes within the CNS where they may serve either pro-inflammatory, or anti-inflammatory, roles.

### 3.6. MiRNAs In Neuroinflammatory Signaling

The cumulative effect of the miR-155, miR-146a, miR-124, miR-21 and let-7 on inflammatory signaling pathways is summarized in Figure 1. The action of multiple miRNAs on these pathways may be either synergistic or antagonistic. For instance, both miR-146a and miR-21 target different components of the TLR/MyD88/NF-κB and JAK-STAT pathways, jointly working to suppress activation of these pathways [44,46,61,64]. Conversely miR-155 promotes activation of the JAK-STAT pathway by targeting the repressors SOCS1 and SHIP1 [36,38]. TLRs are an important component of the neuroinflammatory response and are targeted by miR-124 and let-7, although let-7 may also lead to TLR activation by serving as a DAMP [53,70,72]. The C/EBP family of transcription factors are important for various inflammatory processes such as M1/M2 polarization, and are targeted by miR-124, miR-155 and let-7 [39,51,67]. 

It is interesting to note that miR-155, miR-146a, miR-124 and miR-21 are all induced in response to MyD88/NF-κB dependent signaling [37,42,53,61], implying that they work together to “fine-tune” the neuroinflammatory response. However, the effect of individual miRNAs is cell-type and context dependent as miRNAs may only target transcripts which are actively being expressed. It is likely that these miRNAs have roles in the CNS outside of regulating inflammatory signaling, such as regulating neuronal processes. The presence of miRNAs in extracellular vesicles such as exosomes allows them to participate in intercellular communication [32]. MiR-124, miR-21 and let-7 have been found in exosomes released from neurons, regulating nearby cells such as microglia [74,75,76]. This raises the possibility that even miRNAs expressed in neurons may contribute to inflammatory signaling should they act on neighboring glial cells. Therefore, regulation by miRNA is highly complex, subject to fluctuation and should be carefully assessed in the context of neuroinflammation.

## 4. MiRNAs in Disorders of Neuroinflammation

MiRNAs have been implicated in the pathogenesis of a variety of neurological disorders, consistent with their proven roles in neuroinflammation. Many parallels can be drawn between the mode of regulation by miRNA in such disorders, even among such diverse pathologies as neurodegenerative diseases, viral infections of the CNS and CNS injury.

### 4.1. Neurodegenerative Diseases

#### 4.1.1. Multiple Sclerosis

Multiple sclerosis (MS) is a chronic autoimmune disease of the CNS that most often presents in young adults (20–40 years old), predominately females, and manifests as a neurodegenerative disease caused by demyelination [77]. Although its etiology remains unknown, the neurological symptoms associated with MS arise following impaired neuronal signaling. This is due to the formation of lesions in the nervous system that are associated with the loss of oligodendrocytes and myelin, astrogliosis and neuroinflammation [78]. The neuroinflammatory component of MS results from the infiltration of macrophages and autoreactive lymphocytes into the CNS following disruption of the BBB [78]. Neurodegenerative changes in addition to neuroinflammation promote the characteristic MS pathology and contribute to the progression of the disease [79]. Among the diverse molecular players that regulate neuroinflammatory processes during MS are microRNAs.

Several miRNA inflammatory regulators are overexpressed in the lesions of MS patients including miR-155, miR-146a, miR-21 and miR-326 [80]. Interestingly genetic variants of miR-146a are also associated with MS susceptibility and relapse [81,82]. However, the effects of miR-146a on inflammatory signaling during MS remain controversial: one study reported miR-146a deficient mice had reduced inflammation and demyelination [83] while another reported that delivery of miR-146a mimics to mice increased M2 microglial polarization and remyelination [84]. The activation of microglia to a pro-inflammatory state is considered to be a hallmark of MS pathology [85]. In addition to the pro-inflammatory effects of miR-155 upregulation, the activation of microglia may be mediated via downregulation of the anti-inflammatory miR-124 within microglia. This has been observed in mice with autoimmune encephalomyelitis (an experimental model of MS) [51]. Three microRNAs which are highly upregulated in active MS lesions, miR-34a, miR-155 and miR-326, each target the inhibitor of phagocytosis CD47, which may lead to enhanced myelin phagocytosis by macrophages [80]. MiR-155 may also promote neuroinflammation during MS by targeting key BBB components, leading to an increase in BBB permeability and infiltration of immune cells from the periphery [86]. The polarization of T cells towards different functional subsets is also known to contribute to the immunopathology of MS, and may be influenced by a number of miRNAs including miR-155, miR-146a, miR-326, miR-301a and miR-182 [87]. Therefore, regulation of inflammatory processes by miRNA plays diverse and important roles in the pathogenesis of MS.

#### 4.1.2. Alzheimer’s Disease

Alzheimer’s Disease (AD) is the most common cause of dementia worldwide, and is a chronic and progressive neurodegenerative disorder that results from the loss of synapses and eventually, neuron death [88]. The two hallmarks of AD pathology are the buildup of extracellular plaques of aggregated amyloid beta (Aβ) peptides and intracellular neurofibrillary tangles (NFTs) of hyper-phosphorylated tau [88]. The Aβ plaques originate from the amyloid precursor protein (APP), which is processed into Aβ peptides by a series of enzymatic cleavage steps, including cleavage by β-secretase-1 (BACE-1) [89]. Tau is a microtubule associated protein within neurons that becomes hyperphosphorylated during AD, causing it to aggregate into paired helical filaments (PHFs), that form NFTs within neurons [90]. Neuroinflammation also contributes to much of the pathogenesis seen in AD [91].

MiRNAs are involved in many aspects of Alzheimer’s disease progression including direct regulation of APP expression [92], alternative splicing of APP [93], regulation of BACE [94,95,96], regulation of Tau [97] and even lipid metabolism [98]. Furthermore, miRNA mediated regulation of neuroinflammatory processes also has important implications for AD pathogenesis. TREM2 is a microglial receptor important for mediating clearance of Aβ42 peptides via phagocytosis in the CNS, that is targeted by miR-34a, which is upregulated during AD [99,100]. Both miR-155 and miR-146a are upregulated in the CNS during AD and contribute to regulation of the pathogenic inflammatory signaling that is seen during this disease [101,102]. The expression of miR-146a may be induced by Aβ42 resulting in the attenuation of inflammatory signaling. Therefore it may be acting in a compensatory mechanism during AD [43,103]. The miR-181 family regulates neuroinflammatory signaling in astrocytes [104], and has been shown to be upregulated in an AD mouse model where it interfered with synaptic plasticity through targeting c-Fos and SIRT-1 [105]. The let-7 family are overexpressed and released from neurons during AD, subsequently acting as ligands for TLR7, promoting inflammation and neuronal death [72].

#### 4.1.3. Parkinson’s Disease

Parkinson’s disease (PD) is another neurodegenerative disease of the CNS which mainly causes motor defects due to the death of dopaminergic neurons in the substantia nigra pars compacta, although other neurological symptoms may arise as the disease progresses [106]. Loss of dopaminergic neurons is the central pathological feature of PD, which is accompanied by the accumulation of inclusions within neuronal cell bodies (Lewy bodies) and processes (Lewy neurites), in addition to microgliosis [106]. Lewy bodies are aggregates of insoluble protein, the major component of which is α-synuclein, a protein normally abundant in neurons that misfolds into conformations that confer neurotoxicity during PD [107]. The activation of microglia, which is promoted by α-synuclein, and induction of neuroinflammation are also thought to play a critical role in the pathogenesis of PD [108]. Therefore, it is unsurprising that the influence of miRNA on neuroinflammation plays a role in PD.

Multiple miRNAs, including miR-7, miR-153, miR-34b and miR-34c are known to repress the expression of α-synuclein, and are therefore protective by limiting its associated neurotoxicity and neuroinflammation [109,110]. Furthermore, miR-7 was reported to be downregulated in a mouse model of PD [111], while miR-34b and miR-34c are known to be downregulated in PD brain samples from patients [112]. It follows that downregulation of these miRNAs may result in the overexpression of α-synuclein within neurons during the disease, conferring neurotoxicity and promoting neuroinflammation [111]. The pro-inflammatory miR-155 is upregulated during PD, and plays a central role in the inflammatory response of microglia to α-synuclein [113]. Conversely, overexpression of miR-124 in microglia attenuates the MEKK3/NF-κB inflammatory pathway, limiting the pathogenesis of the disease [114]. 

#### 4.1.4. Prion Diseases

Prion diseases (PrD) are a group of fatal neurodegenerative diseases that arise following misfolding of the cellular prion protein (PrP^C^) into a disease associated conformation (PrP^Sc^) that is uniquely transmissible between individuals [115]. The neurological symptoms associated with the disease are due to loss of neurons, resulting in spongiform change, and are accompanied by deposits of aggregated PrP^Sc^ and gliosis within affected regions of the brain [116]. The central pathological event behind PrD is the self-template directed misfolding of prions, a feature which allows them to propagate throughout the brain and cause widespread neuronal death [117]. Like the other progressive neurodegenerative disorders discussed here, pathological neuroinflammation is central to the progression of PrD [118].

MiR-146a is upregulated in the brain in a variety of different prion diseases, including sCJD [119,120], GSS [120] and rodent models of the disease [121,122,123,124]. Furthermore, miR-146a is induced within microglia in response to signaling by TLRs, which are also overexpressed during prion diseases [125]. Expression of miR-146a within microglia impairs activation towards a pro-inflammatory phenotype, and the overexpression of miR-146a during prion disease may therefore seek to limit excessive neuroinflammation [125]. Many other miRNAs are known to be altered within the brain during PrD, and appear to be dynamically expressed as distinct waves throughout the course of the disease [119,123]. Interestingly, anti-inflammatory miR-124 is upregulated at preclinical stages of the disease, but is downregulated during clinical disease [119,123]. Downregulation of miR-124 may therefore release microglia from quiescence at late stages of the disease and contribute to the excessive neuroinflammation seen during PrDs.

### 4.2. CNS Viral Infection

#### 4.2.1. Japanese Encephalitis Virus

Japanese encephalitis virus (JEV) is an arthropod-borne, neurotropic, positive stranded RNA *Flavivirus* which can cause Japanese encephalitis (JE) in humans, although is usually asymptomatic [126]. JE is a severe disease which can cause death in as many as 30% of cases [127], and is characterized by fever, headache, vomiting and neurological symptoms such as movement disorders, paralysis and seizures [126]. Epidermal tissues are the primary site of JEV infection, following which the virus propagates by infecting circulating blood cells including monocytes and dendritic cells [128]. JEV infiltrates the CNS by crossing the BBB, either through diffusion, receptor mediated endocytosis or possibly through infection of microvascular brain endothelial cells. This ultimately leads to widespread infection of neurons and glia in the CNS and disruption of the BBB [129]. In addition to the neuronal death directly due to JEV infection of neurons, induction of uncontrolled neuroinflammation contributes to the damage within the CNS during JE [128].

miR-155 is upregulated in brain tissue during JEV infection [130] although the role it plays remains debated. One study reported that the miR-155 promoted microglial activation and neuroinflammation by targeting SHIP1 [130], while another study reported overexpression of miR-155 in a human microglial cell line suppressed replication of JEV and impaired innate immune signaling [131]. MiR-146a is another important inflammatory regulator that is upregulated within microglial cells during JEV infection [132,133]. Overexpression of miR-146a during JEV infection limits inflammation by impairing the NF-κB pathway, which also may promote JEV replication via downregulation of interferon-stimulated genes [132,133]. Conversely, the miR-34 family inhibits JEV replication via induction of the type I interferon pathway [134]. JEV can also modulate inflammatory signaling in the host by deregulating miRNAs, including miR-432 [135], miR-301a [136], miR-29b [137], miR-22 [138], miR-370 [139], miR-19b [140] and miR-15b [141]. 

#### 4.2.2. Herpes Simplex Virus Encephalitis

Herpes simplex virus (HSV) normally establishes lytic infection in epithelial cells causing blisters or sores, and maintains a life-long latent reservoir in sensory neurons [142]. Periodic reactivation of HSV within latently infected neurons may lead to recurrent infections, although infection with HSV is usually asymptomatic [143]. Rarely, HSV causes HSV encephalitis (HSVE), which is the most common cause of sporadic encephalitis in humans and has an extremely high mortality rate of about 70% if left untreated [144]. HSVE has symptoms similar to other viral infections of the CNS, including fever, headaches, vomiting, neurological deficits and seizures [145]. Infiltration of immune cells into the CNS, BBB disruption and inflammatory signaling all drive the neuroinflammatory response during HSE, which both promotes clearance of HSV and limits viral replication while at the same time leading to cell death and pathogenic tissue damage, which underlies the severe consequences of HSVE [146].

HSV encodes several miRNAs within its own genome that work in concert with host miR-138, which is highly expressed in neurons, to promote latency by suppressing the expression of viral transcripts that are important during lytic infection [147,148]. In addition to these viral miRNAs, host miRNAs are deregulated during HSVE. For instance, miR-146a is upregulated within neurons during infection with HSV-1, which may contribute to viral evasion of complement by targeting CFH and promote neuropathological changes via activation of elements of the arachidonic acid cascade [149]. HSVE also leads to induction of miR-155 [150,151], which promotes, in this case, protective inflammation as miR-155 deficient mice exhibit increased viral replication and mortality [152]. Both miR-23a and miR-373 suppress the type I interferon (IFN) response by targeting IRF-1, which promotes HSV-1 replication [153,154]. Similarly to JEV, miR-15b was found to be upregulated within the brain during HSVE, hinting at a role for this microRNA in regulating inflammation during multiple viral encephalopathies [141,151]. Multiple members of the miR-200 and miR-182 clusters were found to be upregulated during HSVE, and may play a role in the pathogenesis of the infection by regulating the biosynthesis of heparin sulfate proteoglycans, which are cell entry receptors for HSV [151].

### 4.3. CNS Injury

#### 4.3.1. Ischemic Stroke

Ischemic stroke results from interrupted cerebral blood flow, usually due to an obstruction of blood vessels by blood clots, which results in damage and cell death within the CNS and is often fatal causing over six million deaths each year [155]. Deprivation of blood within the CNS immediately causes tissue damage due to insufficient energy for cellular processes and excitotoxicity, followed by reperfusion which causes edema and BBB disruption [156]. Damage in surrounding regions within the brain is subsequently induced by synaptic loss and the release of pro-apoptotic and pro-inflammatory factors by the dying neurons [157]. Following reperfusion, both microglia and dying neurons release inflammatory mediators concomitant with expression of adhesion molecules by vascular endothelial cells [158]. These changes collectively promote the entry of macrophages and neutrophils into the CNS, which further mediates neuroinflammation [158]. 

MicroRNAs are involved in many aspects of the pathogenic mechanisms that underlie the tissue damage following stroke including excitotoxicity, oxidative stress, BBB damage and apoptosis as well as aspects of post-stroke recovery including neurogenesis and angiogenesis (see [159]). Furthermore, modulation of inflammation by miRNA following ischemia contributes to the progression of pathology. For instance, miR-210 is induced during stroke and appears to promote inflammation by upregulating the expression of pro-inflammatory cytokines and chemokines [160]. Members of the miR-181 family are also modulated following cerebral ischemia and exacerbate tissue damage by promoting neuronal death and inflammatory signaling [161,162,163]. The pro-inflammatory effects of miR-155 are unclear, it has been identified as downregulated following stroke [164], and inhibition of this microRNA limits inflammation and is protective [165,166]. Downregulation of miR-155 may reflect a mechanism to limit inflammation in some regions of the brain following ischemia. Over-expression of miR-124 was shown to be neuroprotective in a mouse model of stroke by promoting M2 microglial polarization and limiting inflammation [167,168]. Similarly, decreased miR-424 expression in the brain tissue of mice following ischemia is protective by suppressing microglial activation and oxidative stress [169,170]. Overexpression of miR-22 was also found to reduce pro-inflammatory cytokines and NF-κB activity while increasing expression of IL-10 and impairing apoptosis [171].

#### 4.3.2. Traumatic Brain Injury

Traumatic brain injury (TBI) is a common cause of death and disability worldwide and may progress towards chronic brain injury [172]. Following primary injury, which results in cell death directly due to physical damage, a heterogeneous group of aberrant biochemical cascades referred to as secondary injury may follow [173]. Secondary injury leads to a series of long-term progressive changes in brain physiology that can cause seizures, sleep disorders or neurodegenerative disorders among others [174]. Several cellular processes mediate secondary injury such as excitotoxicity from increased glutamate, generation of reactive oxygen species, excessive apoptosis and neuroinflammation [174]. Activation of microglia, recruitment of immune cells into the CNS and pro-inflammatory signaling are all central processes that underlie the neuroinflammatory component of TBI [175]. 

One of the best studied miRNAs in TBI is miR-21, which is upregulated in brain microvascular endothelial cells [176] and in extracellular exosomes [177]. miR-21 is considered protective in TBI as it promotes BBB repair and angiogenesis while suppressing apoptosis, and may exert its anti-inflammatory effects on microglia [176,177,178]. miR-107 and miR-103 have been shown to be downregulated in the hippocampus following TBI [179]. These two miRNAs differ by only a single nucleotide and are similarly downregulated during AD [180] implying a related function. miR-107 targets granulin, a protein involved in wound repair, neoplasia and inflammation, and has been implicated in neurodegenerative diseases, therefore suggesting a mechanism by which miR-107 and miR-103 regulate inflammation and neuronal dysfunction [181,182]. The expression of miR-34a is also modulated following TBI, and may influence neuronal differentiation and proliferation by targeting the Notch pathway [183,184]. Furthermore, overexpression of miR-34a along with miR-451 and miR-874 made neurons more vulnerable to injury via increase in pro-inflammatory and pro-apoptotic factors [185]. The pro-inflammatory miR-155 in addition to miR-223 were also significantly overexpressed following TBI, and therefore may play an important role in controlling the response to injury [186]. Similarly to stroke, in vivo inhibition of miR-155 limits neuroinflammation and is protective [187].

### 4.4. Notable miRNAs in Neuroinflammatory Disorders

Some miRNAs are outstanding in that they have been implicated in many of the neuroinflammatory disorders previously discussed and are summarized in Table 1. For instance, the upregulation of miR-155 usually coincides with increased neuroinflammation which results in worsened disease severity [86,102,113,187]. However, in the context of viral encephalopathies such as JEV and HSVE miR-155 expression was protective and suppressed viral replication, likely by promoting clearance of virus from the CNS through its associated inflammatory signaling [131,152]. Conversely miR-146a expression dampens inflammatory signaling in a negative feedback loop. Although this may have positive consequences for neurodegenerative diseases [43,84,125], the severity of viral infection in the CNS is worsened by miR-146a expression through attenuation of interferon stimulated gene (ISG) signaling and increased viral replication [149,151]. MiR-124 is another anti-inflammatory microRNA that is usually protective in neurodegenerative diseases and CNS injury by preventing activation of microglia to a pro-inflammatory phenotype [51,114,121,168]. MiR-181 may play a common role in the pathogenesis of AD and ischemic stroke by promoting neuronal death and inflammatory signaling [105,161,162].

Other miRNAs are notable in that they may have unique or diverging roles in neuroinflammation and are also summarized in Table 1. Although MiR-21 is known to be increased in the lesions of MS patients, a role for it in this context has yet to be determined [80]. Upregulation of miR-21 in the brain may reflect an attempt of tissue repair, as seen following traumatic brain injury [176,177]. The let-7 family represent a unique case as they may promote inflammation by directly acting as a DAMP for TLR-7 during AD [72]. This mode of regulation may not be unique to let-7, as miR-21 has also been shown to work as a TLR ligand [188]. MiR-34 is another example of a miRNA with diverging functions as it promotes macrophage phagocytosis in MS by targeting CD47 yet prevented microglia phagocytotic clearance of Aβ42 in AD by targeting TREM2 [80,99,100]. Furthermore, miR-34 targets α-synuclein but is downregulated during PD which contributes to disease pathogenesis, whereas overexpression of miR-34 worsened traumatic brain injury by promoting apoptosis and inflammation [112,185]. Contrary to this, miR-34 is protective in viral encephalitis by inducing the type I interferon response [134].

## 5. MiRNAs as Biomarkers for Neuroinflammatory Diseases

These neurological disorders all feature pathogenic neuroinflammation and thus often exhibit similar and overlapping symptoms, which complicates diagnosis of a specific neurodegenerative disorder. For example, differential diagnosis of MS, AD and PD relies on a combination of careful clinical assessment, brain imaging techniques and detection of disease specific biomarkers such as misfolded proteins or inflammatory cytokines [189]. Identification of the cause of a viral encephalitis such as JEV or HSVE often relies on antibody or PCR based tests, or CSF blood cultures to detect the infectious agent, although this diagnosis is usually delayed several days which impedes the impact of potential treatments [190]. The diagnosis of TBI can be challenging as these disorders may originate from many different causes and take on different levels of severity [191]. Therefore, improved methods to diagnose and differentiate these diseases are required.

As the expression of miRNAs are commonly altered during disease, they have gained much attention for their potential use as biomarkers. It is important to note that many miRNAs are commonly altered in multiple diseases, for instance miR-155 and miR-146a that regulate inflammation in the disorders discussed previously. Some miRNAs are even commonly altered during a spectrum of many different types of disease. For example, expression of miR-155 and miR-21 are altered during cancer, diabetes and coronary artery disease in addition to neurodegenerative diseases [192]. Therefore, in order to distinguish specific pathologies on the basis of miRNA expression, patterns of multiple miRNAs that are specifically altered during the disease state must be identified.

Multiple studies have identified potential miRNA biomarkers for neuroinflammatory diseases including MS [193], AD [194], PD [194], PrDs [195], JEV [196], ischemic stroke [197] and TBI [198]. The use of miRNA as biomarkers to help diagnose these disorders offers several advantages. Not only are cellular miRNA altered during disease, but circulating miRNA in accessible biofluids such as serum or CSF are also altered and can reflect brain pathologies [192]. MiRNAs are also highly stable molecules in tissues including biofluids, therefore their detection may be more reliable than other types of circulating RNA [199]. Identification of altered miRNA signatures during these neuroinflammatory disorders may lay the groundwork towards developing better diagnostic assays which could help distinguish these often-overlapping diseases at early stages of development. Indeed, many of the inflammatory miRNAs within the CNS discussed previously have already been identified in circulating fluids such as plasma, serum, blood and CSF as biomarkers of neuroinflammatory diseases (Table 2). Therefore, it may be possible to use these miRNAs along with others in non-invasive diagnostic assays for neurological disorders.

## 6. Therapeutic Applications of miRNAs in Neuroinflammatory Disease

By regulating the expression of multiple genes and pathways, miRNAs play a pivotal role in the pathogenesis of many neuroinflammatory disorders. The molecular and cellular processes that underlie the pathogenesis of these disorders are often very complex. Given the central role of miRNAs in regulating molecular cascades during health and disease they may serve as important therapeutic targets. [201]. Indeed, miRNA based therapeutics have been proposed in many complex, difficult to treat disorders such as cancer, heart disease and diabetes in addition to neuroinflammation [202].

The ability to control the expression of miRNA in vivo will serve as the basis for the development for miRNA therapeutics, and as such many new tools have been developed to modulate their expression. For instance, miRNA mimics are small synthetic double stranded miRNA molecules which are processed into functional miRNA, permitting the restoration of a miRNA within a cell [203]. Alternatively, miRNA expression vectors can be used to induce the expression of a specific miRNA within tissues [203]. The activity of a miRNA can also be suppressed, usually through delivery of synthetic sequences complementary to the miRNA which block its binding to endogenous mRNA targets, such as antagomirs, locked nucleic acids anti-miRs and miRNA sponges [204].

One of the main challenges towards applying these technologies in the treatment of disorders in the CNS is the delivery of miRNA-based therapeutics across the BBB. Several promising avenues are being explored which could mediate the delivery of such therapeutics. They include non-viral methods such as lipid-based or polymeric nanoparticle-based delivery systems which work to promote the cellular uptake of miRNA therapeutics [205,206]. Focused ultrasound is another technique which can be used to transiently disrupt the BBB, allowing for targeted delivery of therapeutics [207]. Additionally, viral vectors such as adenovirus vectors and adeno-associated virus (AAV) vectors can be used to induce the expression of a miRNA within the CNS [208]. Therefore, the identification of ideal miRNA therapeutic targets and development of efficient delivery systems to the CNS may allow for improved management and treatment of pathological neuroinflammation.

## 7. Conclusions/Perspectives

Pathological neuroinflammation is a process which underlies multiple CNS disorders. Understanding the mechanisms that lead to these inflammatory processes will undoubtedly inform the development of novel therapeutic and diagnostic strategies which could help alleviate the global burden of these disorders. MiRNAs are among the regulatory molecules capable of modulating neuroinflammatory signaling. Indeed, some miRNAs are common to multiple disorders and have been shown to play central roles in controlling inflammation. For instance, both miR-146a and miR-155 are dysregulated during many of the disorders and extensive studies have shown them to play similar roles across multiple pathologies. Other miRNAs may be disease-specific and it is the cumulative effect of the interactions between multiple miRNAs that influences the outcome of neuroinflammation during disease. MiRNAs are also found circulating in biofluids where they take on specific patterns of altered expression during neuroinflammatory disorders. Therefore, there is much potential in identifying these disease-specific patterns as part of a strategy for diagnosis, prognosis and assessing the effectiveness of therapeutic interventions. Furthermore, inducing anti-inflammatory miRNAs or suppressing pro-inflammatory miRNAs could be a therapeutic strategy to ameliorate tissue damage within the CNS following rampant neuroinflammation. Although progress has been made towards understanding the role of miRNAs in neuroinflammation many disorders remain poorly characterized, and there are a multitude of miRNAs that have roles yet to be determined.

## Figures and Tables

**Figure 1 ncrna-05-00035-f001:**
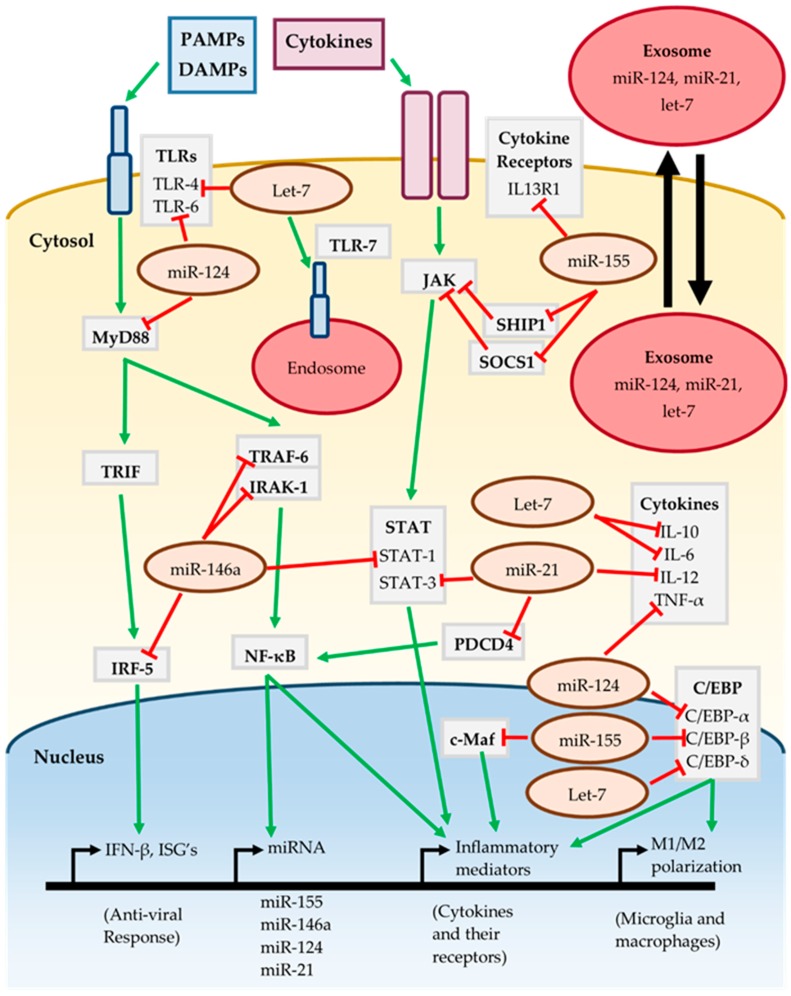
Inflammatory signaling of miRNA. See text for further details and references.

**Table 1 ncrna-05-00035-t001:** Notable miRNAs in disorders of neuroinflammation.

miRNA	Disorder	Expression	Consequences	Reference
miR-155	MS	Up	Increased activation of microglia, enhancement of phagocytosis and increased BBB permeability	[80,86]
	AD	Up	Increased inflammation	[102]
	PD	Up	Microglial inflammatory response	[113]
	JEV	Up	Microglial activation and suppression of viral replication and innate immune signaling	[130,131]
	HSVE	Up	Protective inflammatory response and decreases viral replication	[150,151,152]
	Stroke	Down	Decreased inflammation and tissue damage	[164,165,166]
	TBI	Up	Increased inflammation and tissue damage	[186,187]
miR-146a	MS	Up	Complex, influences inflammation and demyelination/remyelination	[80,83,84]
	AD	Up	Attenuates inflammatory signaling	[43,103]
	PrD	Up	Dampens microglial inflammatory response	[125]
	JEV	Up	Decreased inflammation and ISG secretion, promotes JEV replication	[132,133]
	HSVE	Up	Decreased inflammation, promotes viral replication	[149,151]
miR-124	MS	Down	Release from microglial quiescence	[51]
	PD	-	Attenuates inflammatory signaling in microglia	[114]
	PrD	Up then down	Release from microglial quiescence	[119,123]
	Stroke	-	M2 microglial polarization, decreased inflammation, protective effect	[167,168]
miR-21	MS	Up	CNS specific function unknown	[80]
	TBI	Up	Increased BBB repair and angiogenesis, impaired apoptosis and inflammation	[176,177,178]
Let-7	AD	Up	Acts as a DAMP for TLR-7	[72]
miR-181	AD	Up	Neuronal dysfunction	[105]
	Stroke	Up	Promotes neuronal death and inflammatory signaling	[161,162,163]
miR-34	MS	Up	Enhanced macrophage phagocytosis	[80]
	AD	Up	Impaired Aβ42 clearance by microglia	[99,100]
	PD	Down	Increased α-synuclein expression and increased inflammation	[112]
	JEV	-	Induces type I interferon signaling, decreases viral replication	[134]
	TBI	-	Promotes release of pro-inflammatory and pro-apoptotic factors	[185]

Abbreviations: MS—Multiple Sclerosis; AD—Alzheimer’s disease; PD—Parkinson’s disease; PrD—Prion diseases; JEV—Japanese encephalitis virus; HSVE—Herpes simplex virus encephalitis; TBI—Traumatic brain injury; BBB—blood brain barrier; CNS—central nervous system; ISG—interferon stimulated gene; DAMP—danger associated molecular pattern; TLR—toll like receptor.

**Table 2 ncrna-05-00035-t002:** Inflammatory miRNAs identified as non-invasive biomarkers of neuroinflammatory disorders.

Disorder	Plasma/Serum	Whole Blood	PBMCs	CSF	Reference
MS	let-7	miR-146b	miR-146a, miR-155	miR-181c	[193]
AD	let-7, miR-34, miR-181c, miR-21	let-7, miR-103a, miR-107	miR-34a, miR-181b	miR-146a, miR-155, miR-34a, miR-124, miR-181a	[194,200]
PD	miR-181c	-	-	let-7	[194,200]
PrD	miR-21	-	-	-	[195]
JEV	-	-	-	miR-21, let-7, miR-181a	[196]
Stroke	-	let-7, miR-21	-	-	[197]
TBI	miR-21	-	-	miR-181	[198]

Abbreviations: PBMC – peripheral blood mononuclear cell; CSF – cerebrospinal fluid; MS – Multiple Sclerosis; AD – Alzheimer’s disease; PD – Parkinson’s disease; PrD – Prion diseases; JEV – Japanese encephalitis virus; TBI – Traumatic brain injury.

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
