# Peer review of "MicroRNAs in Neuroinflammation: Implications in Disease Pathogenesis, Biomarker Discovery and Therapeutic Applications"

_ncrna, 2019, doi:10.3390/ncrna5020035_

Round 1

Reviewer 1 Report

Line 59: describe what is meant by 'amoeboid form'

Line 213 and Figure 1: What miRNAs are transported by exosomes?  Please include on your figure the identity of the exosomal miRNAs or remove the reference to exosomes.

Line 255: Missing a word? "...progressive neurodegenerative [disorders?] that results from ...."

Line 310: "...variety of different prion disease[s]..."

Line 315: It is difficult to dissect whether the changes in miRNAs observed during PrD is a result of the disease or contributing to the pathology.  Cause or effect? This sentence makes it seem like it is a one way relationship (i.e., changes in miRNAs-->PrD)

JEV section: You may also want to include (Smith et al; J Virol. 2017 Mar 29;91(8)) which found that miR-34a inhibits JEV replication through activation of interferon signaling

Author Response

Comment: “Line 59: describe what is meant by 'amoeboid form' 

Revision: It was added to the text here that ‘amoeboid’ form indicates the microglia are capable of cell motility and phagocytosis.

Comment: “Line 213 and Figure 1: What miRNAs are transported by exosomes?  Please include on your figure the identity of the exosomal miRNAs or remove the reference to exosomes. 

Revision: miR-124, miR-21 and let-7 are transported in neuronal exosomes to act on microglia; This was specified in the text and figure. 

Comment: “ Line 255: Missing a word? "...progressive neurodegenerative [disorders?] that results from ...."  

Revision: Missing word was added. 

Comment: “Line 310: "...variety of different prion disease[s]..." “ 

Revision: “Disease” was changed to “diseases” 

Comment: “Line 315: It is difficult to dissect whether the changes in miRNAs observed during PrD is a result of the disease or contributing to the pathology.  Cause or effect? This sentence makes it seem like it is a one way relationship (i.e., changes in miRNAs-->PrD) ” 

Revision: The statement “hinting at complex roles in the pathogenesis of PrDs” was removed from this sentence.

Comment: “JEV section: You may also want to include (Smith et al; J Virol. 2017 Mar 29;91(8)) which found that miR-34a inhibits JEV replication through activation of interferon signaling 

Revision: This point was added both to the text and to Table 1. 

Reviewer 2 Report

The document is well written. It resumes the state of the art of miRNAs related to neuroinflammation proces. There is little critics to do. I only have a comment to stress. In line 99 the authors argue that “The genes encoding miRNAs are transcribed by RNA polymerase II”. However, since 2006 it is known that some of them may be transcribed by RNA polymerase III, at least in human (RNA polymerase III transcribes human microRNAs Glen M Borchert, William Lanier & Beverly L Davidson Nature Structural & Molecular Biology volume13, pages1097–1101 (2006)). I think that this comment could improve the text. 

Author Response

Comment: “However, since 2006 it is known that some of them may be transcribed by RNA polymerase III, at least in human (RNA polymerase III transcribes human microRNAs Glen M Borchert, William Lanier & Beverly L Davidson Nature Structural & Molecular Biology volume13, pages1097–1101 (2006)). I think that this comment could improve the text. 

Revision: Transcription of miRNA by RNA pol III was stated in the text.

Reviewer 3 Report

The authors with this review investigated the state of art of miRNAs in pathological neuroinflammation. I find the paper acceptable for the publication.

Major point:

- I think the authors should discuss better the  non-invasive capacity of miRNAs as diagnostic/prognostic tools.  for example in table 1 the authors should add this information specifying if the miRNAs were already found in biofluids (saliva, blood,…)

See: Bertoli G, Cava C, Castiglioni I. The potential of miRNAs for diagnosis,

treatment and monitoring of breast cancer. Scand J Clin Lab Invest Suppl.

2016;245:S34-9. doi: 10.1080/00365513.2016.1208444

Author Response

Comment: “- I think the authors should discuss better the  non-invasive capacity of miRNAs as diagnostic/prognostic tools.  for example in table 1 the authors should add this information specifying if the miRNAs were already found in biofluids (saliva, blood,…) 

Revision: A second table (Table 2) was added that specified which of the inflammatory miRNAs were also identified as biomarkers in biofluids for each of the neuroinflammatory diseases discussed. 

Round 2

Reviewer 3 Report

The paper was improved